# Reversing a reported case of transoceanic dispersal: Nudibranch identifications among tsunami debris

**Katherine O. Montana**[1,2,3]*, **Terrence M. Gosliner**[1,2], **Sarah C. Crews**[1], **Lynn J. Bonomo**[1¤a¤b], **James T. Carlton**[4], **Rebecca F. Johnson**[1,2]

**1** Department of Invertebrate Zoology, Institute for Biodiversity Science and Sustainability, California Academy of Sciences, San Francisco, California, United States of America, **2** Department of Biology, San Francisco State University, San Francisco, California, United States of America, **3** Department of Integrative Biology, University of California, Berkeley, Berkeley, California, United States of America, **4** Coastal and Ocean Studies Program, Williams College-Mystic Seaport, Mystic, Connecticut, United States of America

¤a Current address: Department of Biological Sciences, Northern Arizona University, Flagstaff, AZ, United States of America

¤b Current address: Center for Ecosystem Science and Society, Northern Arizona University, Flagstaff, AZ, United States of America

* kmontana@calacademy.org

**Data Availability Statement:** All data are publicly available via GenBank. GenBank can be accessed via https://www.ncbi.nlm.nih.gov/genbank/. Accession numbers for relevant data hosted on

## Abstract

In the aftermath of the 2011 east Japanese earthquake and tsunami, anthropogenic debris from the east coast of Japan floated across the Pacific Ocean to the west coast of North America. One such vessel from Iwate Prefecture arrived on the coast of Oregon, and the fouling community included specimens identified as the nudibranch *Hermissenda crassicornis*, which was previously thought to range from Japan to Baja California but has since been split into three species: *H. crassicornis* (northeastern Pacific), *H. opalescens* (southeastern Pacific), and *H. emurai* (western Pacific). Also aboard were nudibranchs of the genera *Dendronotus* and *Eubranchus*. Previous work suggested that all of the motile invertebrates found in the tsunami debris fouling community were either pelagic or Japanese in origin. Our study sought to determine whether the *Hermissenda* nudibranch specimens collected from the Iwate vessel were, according to the most updated classification system, only *H. emurai* as would be the case if the nudibranchs were Japanese in origin. In addition, we also sought to identify the *Dendronotus* and *Eubranchus* aboard. Results from DNA sequencing and limited morphological analysis indicate that specimens of *H. crassicornis*, as it is currently recognized, and *H. opalescens* were found on the vessel. Morphological or genetic data resolved the other nudibranchs as the Eastern Pacific *Dendronotus venustus* and *Eubranchus rustyus*. These findings indicate that these species settled after arrival to the west coast of North America. Data shared on GBIF and the iNaturalist platform were also used to map where eastern Pacific *Hermissenda* are currently understood to occur.

GenBank can be found in this manuscript's Supporting Information files via "S1 Table.xlsx".

**Funding:** Our work was supported by the National Science foundation grant DBI-1852276 to Esposito and Johnson, nsf.gov. The NSF did not play a role in the study design, data collection and analysis, decision to publish, or preparation of the manuscript.

**Competing interests:** The authors have declared that no competing interests exist.

## Introduction

Nudibranchs (Nudibranchia Cuvier, 1817) are mollusks of the class Gastropoda that are prevalent in virtually every marine habitat but have been most thoroughly studied in intertidal zones. They are colloquially known as sea slugs and exhibit a wide variety of striking coloration and patterns. Nudibranchs typically have planktonic larvae, and the presence of prey signals them to settle onto the substrate and continue their development [1, 2]. Their prey consists of benthic colonial organisms, including hydroids, tunicates, sponges, and other sessile invertebrates, and they sometimes consume mobile prey, including other nudibranchs [3, 4], or even eggs of other nudibranchs or fishes [2, 5]. With this reproductive strategy and variety of diet possibilities, oceanic currents may disperse nudibranch larvae, and rafts with prey on board may provide opportunity for biofouling. The subject of our study may have faced this scenario. We focus on the three nudibranch species of the genus *Hermissenda* Bergh, 1878, which are found on the coasts of both the western and eastern Pacific in the northern hemisphere. *Hermissenda* was one of the taxa found on human-made debris from the 2011 east Japan earthquake and tsunami that rafted to the coast of Oregon [6].

Following the March 11, 2011 east Japan earthquake and tsunami, also known as the Tōhoku earthquake and tsunami event, thousands of human-made objects, including hundreds of vessels, arrived near or washed up on the shores of the northeastern Pacific Ocean and the Hawaiian Islands. Many of these objects supported marine species native to Japan [6]. One of these vessels, subsequently designated as Japanese Tsunami Marine Debris (JTMD)-BF-356, from Iwate Prefecture was discovered roughly 8 km offshore of Seal Rock, Oregon in April 2015, (Fig 1A and see Methods). More than 20 living species of Japanese invertebrates and fishes were aboard vessel JTMD-BF-356 [6–8]. These included sponges, mussels, crustaceans (amphipods, isopods, and barnacles), bryozoans, the Japanese yellowtail jack *Seriola aureovittata* Temminck and Schlegel, 1845, and the barred knifejaw *Oplegnathus fasciatus* (Temminck and Schlegel 1844). Also aboard were oceanic, neustonic species acquired in transit from Japan to Oregon, including pelagic crabs (*Planes marinus* Rathbun, 1914), barnacles (*Lepas* Linneaus, 1758 sp.), nudibranchs [*Fiona pinnata* (Eschscholtz, 1831)], and bryozoans [*Jellyella tuberculata* (Bosc, 1802)], the latter indicating a colder-water, higher latitude route across the north Pacific. The organisms found on the vessel, and those discovered through the Carlton et al. (2017) study, likely faced effects consistent with marine species invasion, such as occupying artificial habitats and high propagule pressure [9]. These organisms participated in

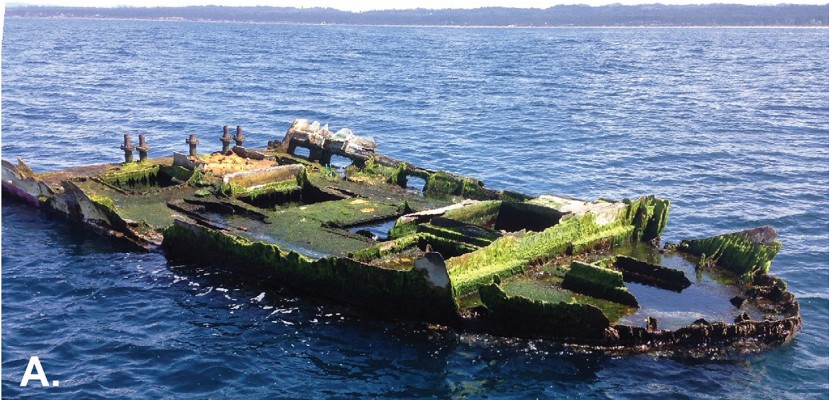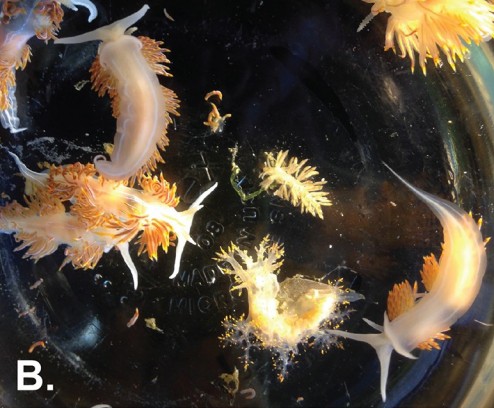

**Fig 1. Vessel JTMD-BF-356 and associated nudibranchs. A.** Vessel JTMD-BF-356 discovered off the coast of Seal Rock, Oregon in April 2015, traced back to Iwate Prefecture, Japan; **B.** Nudibranchs found in a cluster in the vessel. Photographs by John Chapman [7].

the transoceanic spread of non-indigenous marine species which commonly occurs via processes like the exchange of ballast water and hull fouling [10, 11]. Rafting, too, has previously been shown as a mechanism for species dispersal [12–14].

In vessel JTMD-BF-356, researchers and state officials found a cluster of approximately 100 nudibranchs (Fig 1B). Carlton et al. (2017) hypothesized that the nudibranch specimens rafted from Japan as did the rest of the marine organisms found on the debris. Many of the nudibranchs were determined by their external morphology to be *Hermissenda crassicornis* (Eschscholtz, 1831) based on the taxonomy at the time of data analysis. Others were determined to be *Dendronotus* and *Eubranchus* species [6], genera also found in both the eastern and western Pacific [15]. The Carlton et al. (2017) visual identification of *Hermissenda crassicornis* (made by G. McDonald and T. Gosliner previous to the publication of Lindsay and Valdés [2016] [16]), was reasonable when *H. crassicornis* was thought to range across the western Pacific and northeastern Pacific. However, in line with *Hermissenda*'s convoluted taxonomic and systematic history (Fig 2), Lindsay and Valdés (2016) determined the genus to be a complex of three species rather than a single species: *Hermissenda crassicornis*, *H. opalescens* (JG Cooper, 1863), and *H. emurai* (Baba, 1937)—thus bringing the true identities of the JTMD-BF-356 *Hermissenda* into question.

The external morphology of *H. crassicornis* is defined by white lines on their cerata, *H. opalescens* has white tips on their cerata and lacks white lines, and the cerata of *H. emurai* are variable. However, cryptic speciation [26] and generally similar patterning between the species has likely contributed to confusion regarding species delineation. To complicate matters, the already ambiguous species-defining cerata characteristics were lost in the time between photographing the specimens and receiving the specimens, which had all been consolidated for preservation. The nudibranchs' colors had faded to a muted beige, and many of the cerata had fallen off. The only record of their original coloration is from the single photograph in Fig 1B. with no way to discern which specimen corresponded with which nudibranch in the photograph.

Another source of complication in the identification of the nudibranchs present on vessel JTMD-BF-356 arises when we consider the ranges of the species of *Hermissenda*. At the time that the vessel was discovered, all *Hermissenda* from the western to the eastern Pacific were known as *H. crassicornis* [25]. Via "molecular phylogenetics (based on four genes), species delimitation analyses, population genetics, and morphological comparisons," Lindsay and Valdés (2016) separated the species with ranges as follows: *H. crassicornis*, found from Alaska south to Point Reyes, CA; *H. opalescens*, found from Bodega Bay, CA south to the tip of Baja California Sur, Mexico; and *H. emurai*, found in Japan, Korea, and the Russian Far East. Thus, Lindsay and Valdés (2016) reported the area of overlap between the two species to be between Bodega Bay and Point Reyes, CA. Next, additional observational data in the northeastern Pacific extended the range of *H. opalescens* north to Vancouver Island, Canada [27], thereby increasing the area of sympatry between *H. opalescens* and *H. crassicornis*.

Although species delineation, determination of species boundaries, and relationships between species were not the goals of this work, we used phylogenetic, phylogeographic, and population genetic techniques to unravel the mystery of the identity of the specimens from JTMD-BF-356. We determined that simply BLASTing our COI sequences was insufficient to answer our question of identifying the JTMD-BF-356 *Hermissenda*. In addition to confirming the identities of the nudibranchs from the vessel, we wanted to investigate whether all the *Hermissenda* were the same species, and if not, to what species they were each most similar. For the first time, we considered all sequence data from GenBank for *Hermissenda*. These data were added to sequence data from the specimens found on the tsunami debris, along with data

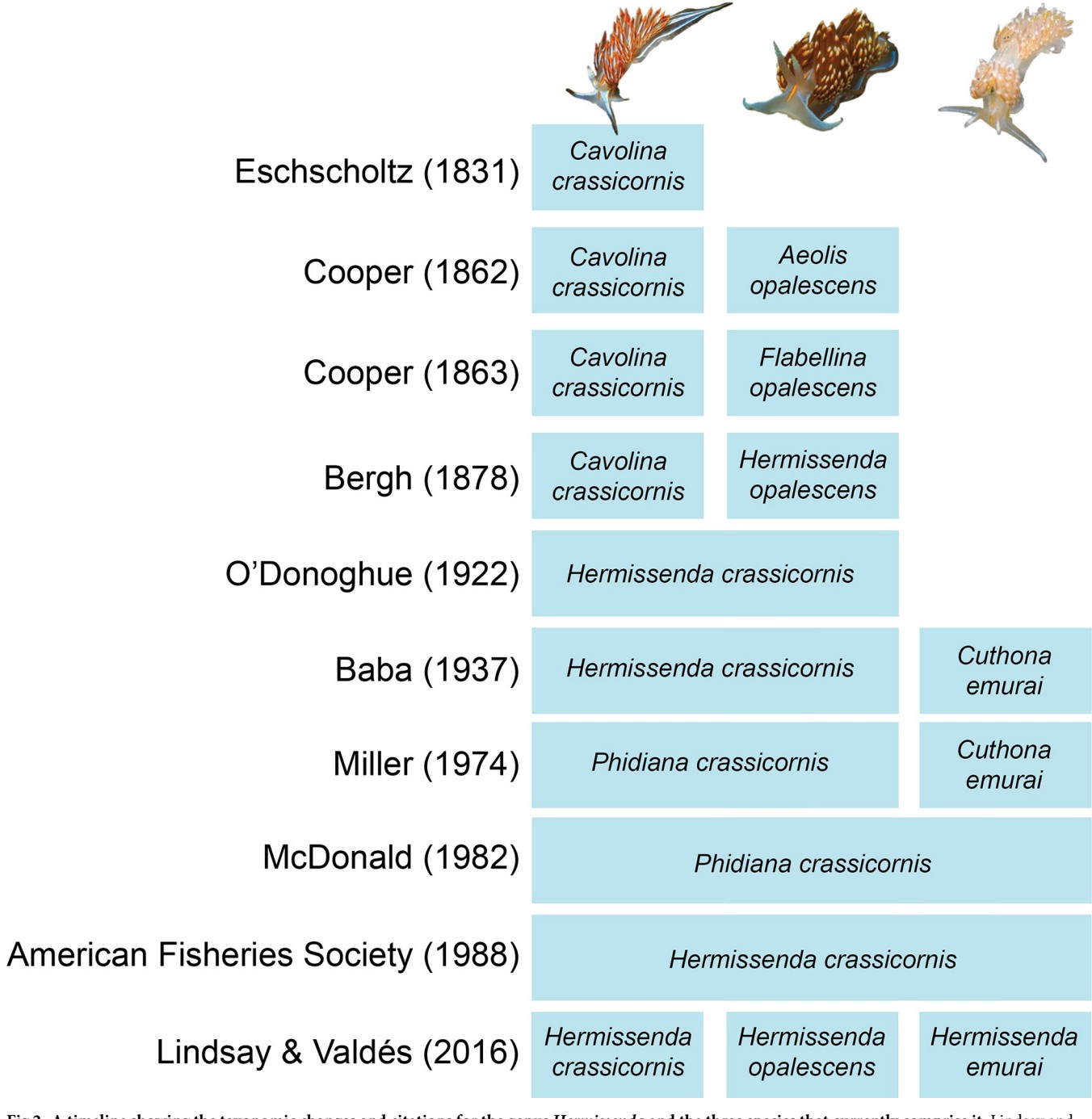

**Fig 2. A timeline showing the taxonomic changes and citations for the genus *Hermissenda* and the three species that currently comprise it.** Lindsay and Valdés (2016) detailed many of the changes that are included [16–25].

from additional coastal Oregon specimens to determine which species were on the debris. By identifying the JTMD-BF-356 nudibranchs in a phylogenetic context, we continue the legacy of *Hermissenda*'s story of taxonomic complexity and connection to geography amidst anthropogenic forces.

## Methods

### Ethics

JTMD collections required no collecting permits. Throughout the JTMD work since 2012, many specimens were submitted by state or provincial officials from Alaska to California and Hawaiʻi.

Additional specimens were collected according to Oregon Department of Fish and Wildlife 2019 Scientific Taking Permit–Fish #22933 scientific collecting permits in the state of Oregon. No ethics review committee was required.

### Vessel JTMD-BF-356

On April 6, 2015, the 7.9 meter-long bow section of a large Japanese fishing vessel was found floating at sea 8 km offshore west of Seal Rock, Oregon, approximately 16 km south of Newport, Oregon [6–8]. The specific origin of the vessel in northern Honshu was likely Iwate Prefecture and left its shores on March 11, 2011, the day of the Tōhoku earthquake and tsunami event [8]. The condition of the vessel suggested that it had been on the seafloor in Japan for several years after the tsunami, and then floated free around 2013–2014; the 1- to 2-year age of Japanese fish found inside the vessel when it landed in Oregon corresponds to this estimate of time-of-departure from Japan.

The vessel was first sampled at sea on April 9, 2015 and then towed to a regional marina and extensively sampled on April 10–11, at which time the nudibranchs noted above were found in a bow internal compartment (which would have been exposed to wash from ocean waves) (S. Rumrill, personal communication, April 2015). The live nudibranchs were then photographed by J. Chapman at the Hadfield Marine Science Center, Oregon State University, Newport, Oregon, who then provided us with the specimens.

### Molecular analyses of nudibranchs

We dissected portions of tissue for DNA extraction from the foot of 45 nudibranch specimens. Thirty-nine were *Hermissenda* from vessel JTMD-BF-356, one was *Eubranchus* sp., and one was *Dendronotus* sp. Four *Hermissenda* were collected on our behalf by N. Treneman in 2019 from Cape Arago, Cape Sebastian, and Cape Ferrelo, OR for comparison to the specimens from the vessel (S1 Table). Extractions were done with the Qiagen DNEasy Blood and Tissue Kit according to the manufacturer's protocol. Cytochrome Oxidase subunit I (COI) was chosen for molecular analysis because it evolves quickly, can elucidate differences between sequences at the species level, and is compatible with existing species delineation data in *Hermissenda* and related groups [16]. The primers used for polymerase chain reaction (PCR) were LCO-1490: 5'-ggtcaacaaatcataaagatattgg-3' and HCO-2198: 5'-taaacttcagggtgaccaaaaaatca-3' [28]. The PCR protocols used were: an initial denaturation for 33 min at 94˚C; 40 cycles of: 94˚C at 30 s for denaturation, an annealing temperature of 46–48˚C for 30 s, and an elongation temperature of 72˚C for 45 s; and a final extension time of 7 min at 72˚C.

In addition to our lab-collected data, we included all existing *Hermissenda* COI sequences on GenBank (S2 Table), as well as several closely related species as outgroups to root our tree: *Godiva quadricolor* (Barnard, 1927), *Phyllodesmium jakobsenae* Burghardt and Wägele, 2004, *Phidiana lascrucensis* Bertsch and AJ Ferreira, 1974, *Nanuca sebastiani* Er. Marcus, 1957, and *Dondice occidentalis* (Engel, 1925). Sequences were aligned using Muscle 2.8.31 [29] with default settings built into Aliview version 1.26 [30]. Gene trees were constructed using IQTree 2 for maximum likelihood and MrBayes 3.2 for Bayesian inference [31, 32]. Model selection was performed by ModelFinder within the IQTree web platform [33, 34], and the maximum

likelihood analysis was partitioned by codon position according to the following models: codon 1:TNe+I, codon 2: F81+F, codon 3: TN+F+G4 [35]. The Bayesian tree was constructed using equivalent models based off of the ModelFinder results: codon 1: SYM + I, codon 2: F81, codon 3: SYM + gamma. Ten thousand ultrafast bootstraps were done in IQTree [36], and MrBayes was run for 10 million generations, saving every 1000th generation. We built a median joining haplotype network (epsilon = 0) using PopART to further illustrate the relationships between our specimens and existing GenBank sequences [37]. As a supplement, we examined species delimitation using the automatic barcode gap discovery website (ABGD) under Jukes Cantor, Kimura K80, and simple distance criteria, with X = 1.5, Pmin = 0.001, Pmax = 0.1, steps = 10, and Nb bins = 20 [38]. Also supplementary, we ran a coalescent-based ASTRAL tree estimation [39].

Scanning electron microscopy (SEM) was used to image the radulae to observe minute morphological differences between a single specimen each of *Hermissenda opalescens* and *H. crassicornis* from vessel JTMD-BF-356; species identity was determined from the phylogenetic analyses of the COI data. The SEM samples were coated with gold/palladium using a Cressington 108 Auto vacuum sputter coater, and micrographs were taken using an Hitachi SU3500 scanning electron microscope at the California Academy of Sciences.

We mapped occurrence data from GBIF (including Research Grade community-collected data from iNaturalist) for *Hermissenda opalescens* and *H. crassicornis* along the west coast of North America using the R packages sf v.1.0.9, ggplot2 v.3.4.0, tidyverse v.1.3.2 [40–44].

## Results

### Molecular analyses and phylogenetic relationships of *Hermissenda*

We combined the 658-bp COI region from 43 *Hermissenda* specimens with 62 COI sequences from GenBank for phylogenetic analysis. The topology of the gene trees for the Bayesian and fast maximum likelihood analyses showed the same support values and relationships (Fig 3). There was 100% support for the *Hermissenda* clade in both analyses, with *H. opalescens* sister to (*H. emurai*+*H. crassicornis*). Low support (<95/0.95) characterized the *H. opalescens* clade across all support metrics. The *H. emurai* clade had high Bayesian support (0.99) but fared lower via SH-aLRT and UFBoot supports (80.2 and 79, respectively). The *H. crassicornis* clade had a high SH-aLRT value (92.6) but low Bayes and UFBoot support (0.51 and 89, respectively). The clade containing *H. emurai* and *H. crassicornis* sister to one another recovered low support. A coalescent-based estimation of intraspecific relationships can be found in the ASTRAL phylogeny, with relatively low (0.75) support for the *H. opalescens* and (*H. emurai*+*H. crassicornis*) clades but high support (0.95) for *Hermissenda*. (S1 Fig). Three species groups that correspond to the three species (*H. crassicornis*, *H. emurai*, and *H. opalescens*) were recovered from the ABGD analysis (S3 Table and Fig 3). The specimens from vessel JTMD-BF-356 were recovered in the *H. crassicornis* and *H. opalescens* clades, and none were found in the *H. emurai* clade.

Previous work included haplotype networks for each *Hermissenda* species separately, but none have shown all three together [16, 27]. The haplotype network we produced shows three distinct groups, with our samples clustering according to species (Fig 4). The network also shows three COI mutations present between the closest *H. crassicornis* and *H. emurai* specimens. There are 11 mutations between the closest *H. emurai* and *H. opalescens* specimens. The most distantly related *H. emurai* have seven mutations between each other. The most frequently occurring haplotype falls within the *H. crassicornis* cluster. Within the haplotype network the specimens from vessel JTMD-BF-356 correspond to *H. crassicornis* and *H. opalescens*, but not *H. emurai*.

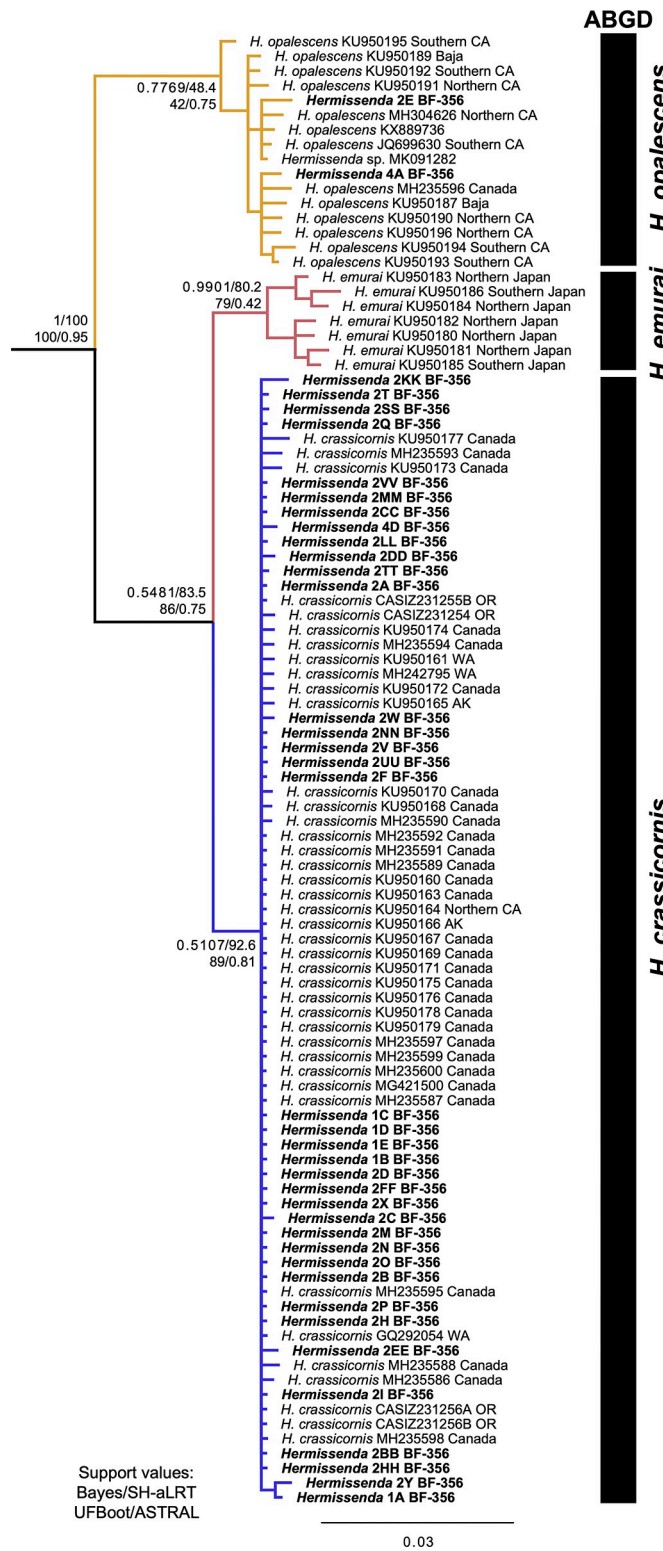

**Fig 3. Phylogeny showing relationships of *Hermissenda* species based on COI.** Support values (Bayes/SH-aLRT/ UFBoot) are shown for the nodes that differentiate the species. Intraspecific relationships between specimens are less central to this study, and, thus, their support values are not included in Fig 3. ABGD results recovered three groups, one for each species. Samples from vessel JTMD-BF-356 shown in bold. Numbers associated with JTMD-BF-356 specimen names indicate the jar in which they were initially stored. Other terminal labels either have GenBank

numbers and their locations, if known (state abbreviations included), or a California Academy of Sciences catalog collections number to indicate that they were new samples collected from Oregon in 2019.

## Other results

The SEM images of the radulae of the two specimens on the JTMD-BF-356 vessel showed serrations under the central cusp with denticles on either side of the cusp, typical of *Hermissenda*. *Hermissenda crassicornis* had five denticles from the central cusp, and *H. opalescens* had four denticles, with no other observable differences on the radulae (Fig 5). While *H. emurai* specimens were unavailable for morphological examination, they typically have 6–7 denticles [16].

Regarding the age of the nudibranchs, the larger *Hermissenda* specimens were approximately 2.5 cm in length. Lab studies of reared nudibranchs found that the largest individuals of *H. crassicornis* from Monterey Bay, CA grew at a rate of 0.68 mm per day post settlement with an approximate 35 day veliger stage [45]. According to that rate, the 2.5-cm *Hermissenda* on vessel JTMD-BF-356 would be approximately 72 days post-hatching.

From their external morphology, the non-*Hermissenda* specimens resemble *Dendronotus venustus* MacFarland, 1966, which was reported as *Dendronotus frondosus* (Ascanius, 1774) in Carlton et al. (2017), a species also regarded at the time as being amphipacific [46]. Carlton et al. (2017) overlooked that Eastern Pacific *D. "venustus"* should now be regarded as the native *D. venustus* [46]. *Eubranchus rustyus* (Er. Marcus, 1961) [reported as *Eubranchus* sp. in Carlton et al. (2017)], both known only from the eastern Pacific. The *Dendronotus* sequencing failed, and the *Eubranchus* specimen was indicated to be *E. rustyus* by a BLAST search of a COI sequence that we produced but that was not included in our phylogeny.

The map using GBIF data shows broad overlap between *Hermissenda crassicornis* and *H. opalescens* (Fig 6). *Hermissenda crassicornis* occurrences range from Alaska south to the southernmost coast of California. *H. opalescens* occurrences range from Vancouver Island, Canada south to Baja California Sur, Mexico.

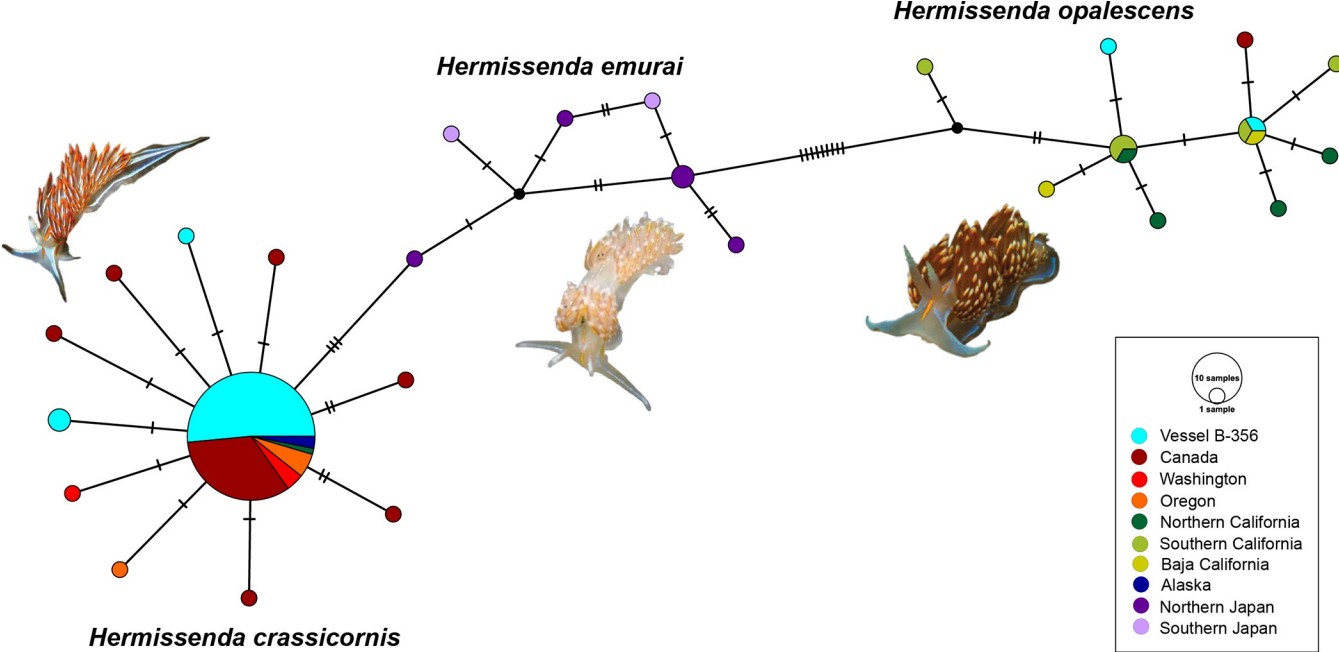

**Fig 4. Haplotype network showing the numbers of mutations between the COI sequences and the geographic origin of each specimen.**

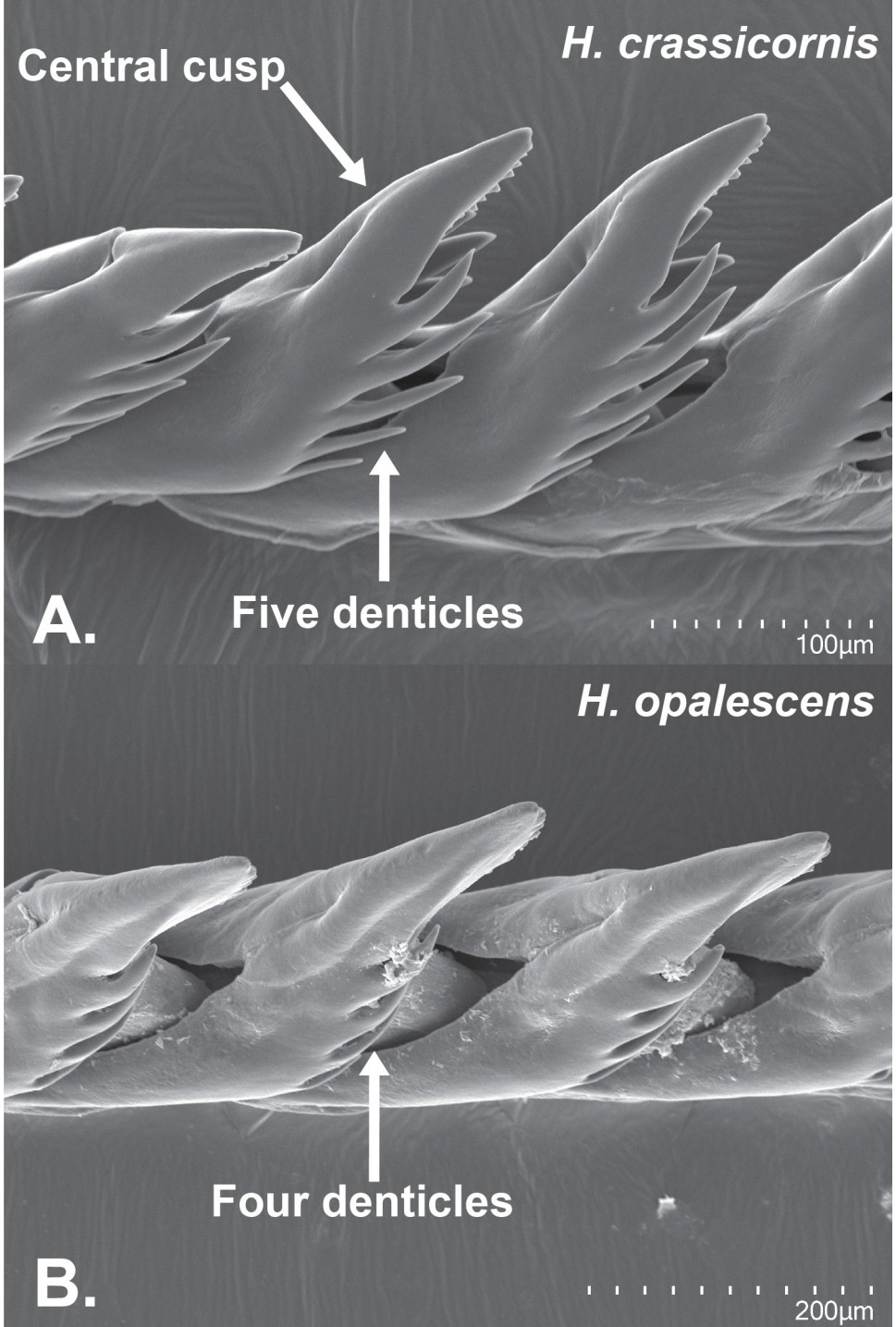

**Fig 5.** Scanning electron microscopy (SEM) images of A. *Hermissenda crassicornis* and B. *Hermissenda opalescens* radulae.

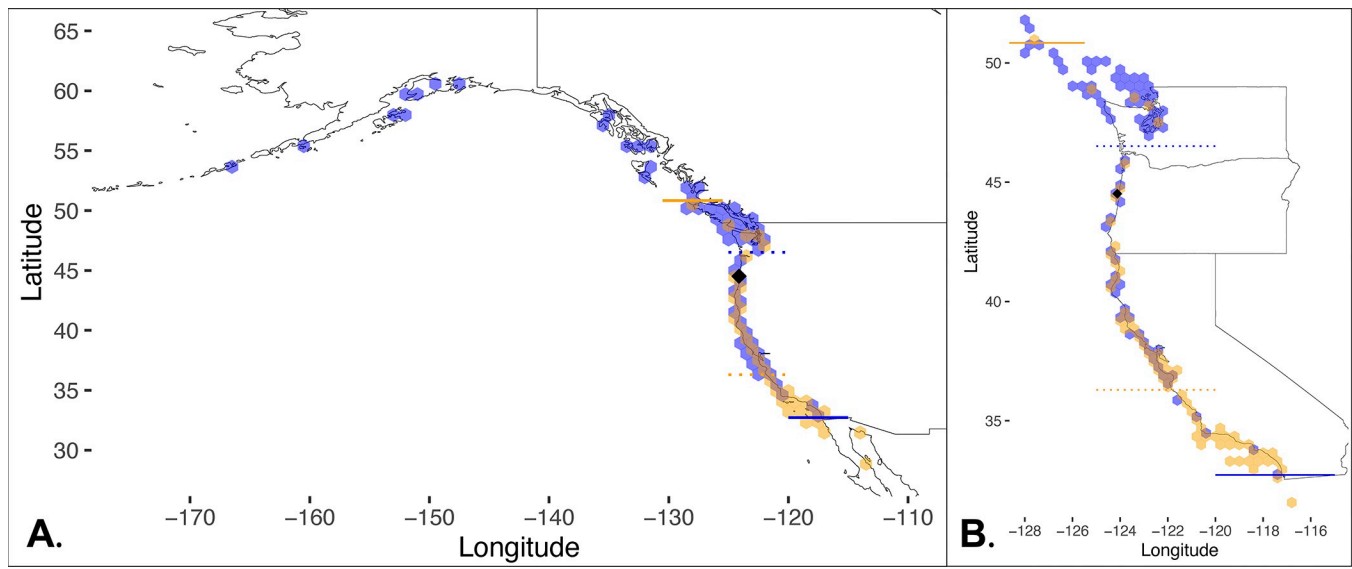

**Fig 6. Maps made from GBIF data showing the area of overlap between *Hermissenda crassicornis* (blue) and *H. opalescens* (orange) occurrences.** The black diamond represents the site offshore of Seal Rock, Oregon where vessel BF-356 was found. A. Shows the entire range of both species, and B. shows a closer view of the west coast of the United States to highlight the overlap. The solid orange line represents the northernmost occurrence of *H. opalescens*, and the dotted orange line represents the mean latitude of *H. opalescens*. The solid blue line represents the southernmost occurrence of *H. crassicornis*, and the dotted blue line represents the mean latitude of *H. crassicornis*. Background map is included in the R package ggplot2 v.3.4.0 [41].

## Discussion

Our study revealed that the nudibranchs on vessel JTMD-BF-356 were *Hermissenda crassicornis* and *H. opalescens*, *Dendronotus venustus*, and *Eubranchus rustyus*. All four species are native to the Eastern Pacific Ocean and occur from Alaska or British Columbia south to California or Baja California [15]. Thus, all four species settled on the vessel after its arrival in North American waters. While the vessel was first intercepted drifting off the central Oregon coast, we do not know its history prior to detection. Large rafted JTMD objects were detected to be drifting along the Oregon and Washington coasts after acquisition east of coastal boundary currents both from the north and the south (JTC, personal observations). However, as noted earlier, the cold water neustonic bryozoan *Jellyella tuberculata* was on this vessel, suggesting that this rafting wreck traveled across the top of the North Pacific and was likely acquired in the region of British Columbia-Washington before then being transported south on the California Current. Given the proposed age (above) of the nudibranchs aboard, larvae of *Hermissenda*, and perhaps the other nudibranchs, settled in the boat in northern waters around January 2015, permitting them to grow to the sizes at which they were found by early April 2015. To our knowledge, this is only the second report of rafting *Hermissenda* in the open ocean, following an earlier report of *Hermissenda* "*crassicornis*" found rafting on kelp [*Macrocystis pyrifera* (Linnaeus C. Agardh, 1820)] in southern California [47]. Bushing further noted that adult *Hermissenda* "were observed laying eggs on the kelp raft that could later hatch, releasing pelagic stages into the open water" [47]. We know of no previous reports of *Dendronotus* or *Eubranchus* rafting in ocean waters.

The addition of the JTMD-BF-356 specimens and 2019 Oregon specimens to existing Gen-Bank COI data for *Hermissenda* provides the most up-to-date molecular phylogeny of the formerly known *Hermissenda crassicornis* species complex COI sequences and supports the currently understood species limits and relationships between *Hermissenda* species [16]. Although the support values on our phylogeny were not high, the haplotype network is

another line of evidence for the JTMD-BF-356 specimens being *H. crassicornis* and *H. opalescens* and maintaining the Lindsay and Valdés (2016) species relationships. However, we do note that even within each group, there is considerable genetic variation, with the most distantly related *H. emurai* being seven mutations away from each other. The ABGD test we ran also supported the existence of three species that correspond to *H. crassicornis*, *H. opalescens*, and *H. emurai* (S3 Table, with p-distance values, and Fig 3). Additional variation within species can be seen in the ASTRAL phylogeny (S1 Fig). Thus, our molecular phylogenetic results, haplotype network, and species delimitation analysis support current *Hermissenda* species limits and relationships and our conclusion that the *Hermissenda* on vessel JTMD-BF-356 did not raft from Japan.

Because we only sampled one specimen each of *H. crassicornis* and *H. opalescens* for SEM imaging, we cannot draw any solid conclusions from those data. *Hermissenda crassicornis* having five denticles coming from the central cusp and *H. opalescens* having four denticles may demonstrate characteristic differences between the two species, but they also may simply be due to intraspecific variation. More sampling is required to determine whether denticle number is a reliable character to differentiate between the two species.

Our GBIF map, including iNaturalist data, agrees with the Merlo et al. (2018) and Goddard et al (2018) [48] range extension of *Hermissenda opalescens* north to Vancouver Island. Our map also shows a more southern extension of *H. crassicornis* than previously understood, extending the area of overlap between *H. opalescens* and *H. crassicornis*. We note that we used only Research Grade observations from iNaturalist, which are fairly reliable, but potential misidentifications may have occurred especially considering the morphological ambiguity between *Hermissenda* species. In fact, we ended up culling the most extreme north samples of *H. opalescens* due to vague cerata morphology in the iNaturalist photographs that could have confused one species for another. Depending on the duration of this area of range overlap, *H. opalescens* could potentially be moving north in response to warming ocean temperatures. There is substantial evidence of poleward shifts of northern range limits in eastern Pacific nudibranchs [48–50].

While dietary strategies were not the focus of this study, we note that the presence of certain food may have played a role in the presence of *H. opalescens* and *H. crassicornis* on vessel JTMD-BF-356. While *H. crassicornis* prefers indigenous food sources to non-indigenous food sources [51], it is possible that the local *Hermissenda* larvae in Oregon settled on vessel JTMD-BF-356 at the cue of the non-indigenous Japanese food sources found on the vessel. Species of *Hermissenda* are known to eat many sessile invertebrates including tunicates, hydroids, and anemones, and specifically *Hermissenda* is known to settle in the presence of and feed on hydroids in the genus *Obelia* Péron and Lesueur, 1810 which has members native to both Japan and North America [3, 52, 53]. *Obelia longissima* (Pallas, 1766), a known prey item of *H. crassicornis*, was found on vessel JTMD-BF-356 [3, 54] and could have potentially served as a food source and settling cue to both *H. crassicornis* and *H. opalescens*; chemicals associated with nudibranch food are often cues for planktonic nudibranch larvae to settle [55]. One possible scenario is that the planktonic larvae settled and were cued to metamorphize by the hydroid (or other) prey found on vessel JTMD-BF-356.

While our taxonomic conclusions are compelling, we note that the discovery of adult nudibranchs on Japanese tsunami marine debris leaves us with two interesting puzzles when considering all of the other 633 (of 634) biofouled sampled JTMD objects analyzed [6]. First, no other JTMD object studied between 2012 and 2017 supported adult invertebrates from the northeast Pacific. Instead, recruits of native northeast Pacific species consisted, without exception, of 1–2 mm nepionic barnacles (such as *Pollicipes polymerus* Sowerby, 1833 and *Balanus glandula* Darwin, 1854) and nepionic bivalves [such as *Crassadoma gigantea* (J.E. Gray, 1825)

and *Kellia suborbicularis* (Montagu, 1803)] [6]. Second, given that JTMD-BF-356 was colonized by no fewer than four species of eastern Pacific nudibranchs, why were no other adult individuals of other species unquestionably native to the eastern Pacific found on this raft? We can only tentatively conclude that an unusual series of events occurred, such that a cohort of *Hermissenda* larvae were washed into the vessel at approximately the same time (almost all of the *Hermissenda* were approximately the same size; J. W. Chapman, pers. comm.). At the same time, or perhaps later, they were joined by *Dendronotus* and *Eubranchus*. Why other eastern Pacific species failed to colonize the vessel we cannot know at this time.

## Conclusions

The four species of nudibranchs from JTMD-BF-356 are eastern Pacific species that settled on the vessel after it arrived on the North American coast. Recruitment of the *Hermissenda* occurred somewhere north of Oregon, given their age. Our results preserve the currently understood species limits and evolutionary relationships of *Hermissenda* species. The addition of iNaturalist and GBIF data also demonstrate more southern occurrences of *H. crassicornis* than were previously expected given their previously published ranges. While many of the species previously studied on JTMD-BF-356 were indicative of transport of northwestern Pacific species to the eastern Pacific, we have shown four cases in which that paradigm has reversed with indigenous colonization of nonindigenous substrate.

## Supporting information

**S1 Fig. ASTRAL phylogeny showing relationships both between and within species of *Hermissenda*, also based on COI.**
(PDF)

**S1 Table. Data for all the COI sequences generated in this study.**
(XLSX)

**S2 Table. Data for the COI sequences downloaded from GenBank.**
(XLSX)

**S3 Table. The Jukes Cantor p-distances between each species from ABGD analysis.**
(DOCX)

## Acknowledgments

We would like to thank L. Esposito for their mentorship of author K. Montana. A. Lam provided training and insights into working in the laboratory and working with haplotype networks. Support in coding for R was provided by G. Rapacciuolo, R. Tarvin, and S. Jacobs. A. Young and A. Miller offered lab moral support. B. Cruz also helped in the lab. We thank S. Rumrill for collecting the nudibranchs from JTMD-BF-356, J. Chapman for photographs of the vessel and the nudibranchs, and G. McDonald and J. Miller for early identification assistance and for further information respectively. We are grateful to N. Treneman for collecting fresh material of *Hermissenda* for us in 2019 from the Oregon coast. C. Piotrowski, L. Kools, and J. Loacker managed our samples in the invertebrate zoology collections at the California Academy of Sciences. We also thank the fellow interns in the Summer Systematics Institute REU program for community and support throughout the project. We thank the nudibranchs for giving their lives to science. We thank the Ramaytush Ohlone and the Confederated Tribes of Siletz Indians for stewarding the land and coasts where our work took place. We thank the anonymous reviewers for their helpful comments.

## Author Contributions

**Conceptualization:** Katherine O. Montana, Rebecca F. Johnson.

**Data curation:** Katherine O. Montana.

**Formal analysis:** Katherine O. Montana, Sarah C. Crews, Lynn J. Bonomo, Rebecca F. Johnson.

**Funding acquisition:** Rebecca F. Johnson.

**Investigation:** Katherine O. Montana, Terrence M. Gosliner, Sarah C. Crews, Lynn J. Bonomo, Rebecca F. Johnson.

**Methodology:** Katherine O. Montana, Terrence M. Gosliner, Sarah C. Crews, Rebecca F. Johnson.

**Project administration:** Rebecca F. Johnson.

**Resources:** Rebecca F. Johnson.

**Supervision:** Rebecca F. Johnson.

**Visualization:** Katherine O. Montana, Rebecca F. Johnson.

**Writing – original draft:** Katherine O. Montana, Rebecca F. Johnson.

**Writing – review & editing:** Katherine O. Montana, Terrence M. Gosliner, Sarah C. Crews, Lynn J. Bonomo, James T. Carlton, Rebecca F. Johnson.

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
