## [Decision Letter · Decision Letter 0]

23 Jul 2024

PONE-D-24-25192Why didn’t the nudibranch cross the ocean? Understanding biogeographic and evolutionary relationships of Hermissenda (Nudibranchia: Myrrhinidae) Bergh, 1878PLOS ONE

Dear Dr. Montana, Thank you for submitting your manuscript to PLOS ONE. After careful consideration, we feel that it has merit but does not fully meet PLOS ONE’s publication criteria as it currently stands. Therefore, we invite you to submit a revised version of the manuscript that addresses the points raised during the review process. Please submit your revised manuscript by Sep 06 2024 11:59PM. If you will need more time than this to complete your revisions, please reply to this message or contact the journal office at plosone@plos.org. Please include the following items when submitting your revised manuscript:A rebuttal letter that responds to each point raised by the academic editor and reviewer(s). You should upload this letter as a separate file labeled 'Response to Reviewers'.A marked-up copy of your manuscript that highlights changes made to the original version. You should upload this as a separate file labeled 'Revised Manuscript with Track Changes'.An unmarked version of your revised paper without tracked changes. You should upload this as a separate file labeled 'Manuscript'.If applicable, we recommend that you deposit your laboratory protocols in protocols.io to enhance the reproducibility of your results. Protocols.io assigns your protocol its own identifier (DOI) so that it can be cited independently in the future. For instructions see: https://journals.plos.org/plosone/s/submission-guidelines#loc-laboratory-protocols. Additionally, PLOS ONE offers an option for publishing peer-reviewed Lab Protocol articles, which describe protocols hosted on protocols.io. Read more information on sharing protocols at https://plos.org/protocols?utm_medium=editorial-email&utm_source=authorletters&utm_campaign=protocols.

We look forward to receiving your revised manuscript.

Kind regards,

Luisa Borges, PhD

Academic Editor

PLOS ONE

Journal Requirements:

5. We note that Figure 2 in your submission contain copyrighted images. All PLOS content is published under the Creative Commons Attribution License (CC BY 4.0), which means that the manuscript, images, and Supporting Information files will be freely available online, and any third party is permitted to access, download, copy, distribute, and use these materials in any way, even commercially, with proper attribution. For more information, see our copyright guidelines: http://journals.plos.org/plosone/s/licenses-and-copyright.

6. We note that Figure 7 in your submission contain map/satellite images which may be copyrighted. All PLOS content is published under the Creative Commons Attribution License (CC BY 4.0), which means that the manuscript, images, and Supporting Information files will be freely available online, and any third party is permitted to access, download, copy, distribute, and use these materials in any way, even commercially, with proper attribution. For these reasons, we cannot publish previously copyrighted maps or satellite images created using proprietary data, such as Google software (Google Maps, Street View, and Earth). For more information, see our copyright guidelines: http://journals.plos.org/plosone/s/licenses-and-copyright.

We require you to either (1) present written permission from the copyright holder to publish these figures specifically under the CC BY 4.0 license, or (b) remove the figures from your submission:

a. You may seek permission from the original copyright holder of Figure 7 to publish the content specifically under the CC BY 4.0 license.  

7. Please remove your figures from within your manuscript file, leaving only the individual TIFF/EPS image files, uploaded separately. These will be automatically included in the reviewers’ PDF.

Reviewers' comments:

Reviewer's Responses to Questions

**Comments to the Author**

1. Is the manuscript technically sound, and do the data support the conclusions?

Reviewer #1: Partly

Reviewer #2: Yes

Reviewer #3: Yes

2. Has the statistical analysis been performed appropriately and rigorously? 

Reviewer #1: N/A

Reviewer #2: Yes

Reviewer #3: N/A

3. Have the authors made all data underlying the findings in their manuscript fully available?

Reviewer #1: Yes

Reviewer #2: Yes

Reviewer #3: Yes

4. Is the manuscript presented in an intelligible fashion and written in standard English?

Reviewer #1: Yes

Reviewer #2: Yes

Reviewer #3: Yes

5. Review Comments to the Author

Reviewer #1: The study reports the species identification for a series of Hermissenda specimens found on debris that crossed the Pacific following the 2011 tsunami in Japan. The complexities regarding the three pseudocryptic species of Hermissenda and the fact that all adults of species identified on tsunami debris to date have originated in the eastern Pacific add a layer of interest to confirming the species IDs. The findings are clear cut: the specimens seem to be eastern Pacific species. Unfortunately, the manuscript provides a rather convoluted and disjointed presentation of this finding. Moreover, related investigations with regard to phylogenetic analyses of the three species and range limits based on iNaturalist could have some improvements and be better incorporated into single narrative for the whole manuscript.

Major Comments

1. Overall, I suggest that the manuscript could be considerably shortened and the flow of ideas made considerably simpler. Currently, the manuscript repeatedly raises the possibility that the Hermissenda could have rafted from Japan, when no evidence supports this, there are several unnecessary tangential paragraphs or sections, and there could be better separation of material between introduction, methods, results and discussion. I suggest the following outline for the manuscript, which I think we would be clearer: Hermissenda was found on debris from Japan. Because of the timing of when they were identified and origin of the debris, they could be any one of the 3 pseudocryptic species. This raises three questions. First, what species were they? Second, since only sequence data are available for IDing the specimens, does the addition of these sequences to molecular phylogenies causes any issues with the clear molecular separation of the 3 pseudocryptic Hermissenda species? Lastly, since it turns out both opalescens and crassicornis are present on the debris, what is the latest evidence on the ranges of those species? Since there is plenty of overlap in range, and these nudibranchs are fast growing generalist feeders, it seems quite feasible that planktonic larvae of both species would have been able to settle and grow to adulthood in the biofouling community on the debris while it was in eastern Pacific waters. All of the various bits for this outline are currently in the manuscript, but they are not laid out in this order and are interrupted by intervening tangential ideas. I suggest revising the entire manuscript to be organized around this outline, starting with a revised title, which could be far more informative of what the major finding is. Suggestion: A trans-Pacific rafting community may have received Hermissenda immigrants from the destination habitat.

2. At the moment, there seems to be some circularities in the way the data is considered. The new specimens from the vessel debris are from an unknown population location. No morphological data on the cerata are presented. Thus, they can only be identified using COI sequence similarity with previously identified specimens from other studies, with no other corroborating evidence (see below re radula dentition). Given this logic flow, it is tautological to then consider the species delineations according to sequences. Proper evidence for that task requires morphological and sequence data for all individuals. I suggest interpretation of the phylogenetic trees and haplotype networks beyond IDing the specimens should be limited to assessing whether the new sequences cause any meaningful change to the clades that correspond to each of the species, as shown in Lindsay and Valdes. This approach may have been the intention from the outset, but that is not how the explanations come across in lines 282-290 or so. Instead the arguments appear to be taking the sequences as evidence supporting species delineation – which is a circular argument since the species IDs were determined using those sequences.

3. No external morphological data is presented in the results. Yet in the discussion, photographs consistent with crassicornis are mentioned. This is highly problematic, both because these were apparently not treated as useful results, but also because it contradicts the molecular identification of some specimens as opalescens. The one picture provided in the methods does not have adequate resolution to assess white tips or lines on the cerata. Surely this data should be thoroughly addressed in the results, and the discrepancy needs to be discussed? At least a couple of the specimens in the methods figure appear to have widely separated groups of cerata, possibly consistent with the morphology of emurai? Note that depending on how this aspect is dealt with it may well change how the molecular data needs to be interpreted. In particular, which sequences in the phylogenetic trees that are assigned to the opalescens clade have pictures showing white stripes? Do they cluster together? Does removing these ambiguous specimens change the node support values?

4. For the range limits, I found the exclusive use of iNaturalist data with no quality control to be troublesome. I am most familiar with the northward end of the range, and a quick check revealed that the 2 most northerly locations for opalescens (Alaska, Haida Gwaii) were a single specimen photographed 4 times that does seem to have white tips and a specimen that could easily be a Dendronotus not a Hermissenda. Tellingly, the Haida Gwaii specimen is not shown in Fig 7, and was changed to Dendronotus following my query about its veracity! Moving a northward range boundary from Vancouver Island (where there are many iNaturalist observations and the evidence from Merlo et al) to Alaska based on such scant and dubious evidence seems to far over reach usual standards. I suggest some thorough checking of the images from the extremes of the distributions, and some systematic sub sampling of all the iNaturalist images to have some idea of the quality of that data. A quick search revealed a range of studies that can give guidance on appropriate methods here. Regardless, I suggest much more cautious language should be used regarding the range limits of the species.

5. It is surprising that the manuscript does not engage with the studies of range shifts among nudibranchs and other taxa along the Pacific coast of North America. In particular, some discussion of whether shifts thought to be associated with marine heat wave ten years ago may be persisting. Some relevant literature (and there may be others – I suggest cross referencing from these studies):

Goddard, J., Treneman, N., Pence, W., Mason, D., Dobry, P., Green, B., and Hoover, C. 2016. Nudibranch range shifts associated with the 2014 warm anomaly in the Northeast Pacific. Bulletin of the Southern California Academy of Sciences 115(1): 15–40.

Goddard, J.H.R., Treneman, N., Prestholdt, T., Hoover, C., Green, B., Pence, W.E., Mason, D.E., Dobry, P., Sones, J.L., Sanford, E., Agarwal, R., McDonald, G.R., Johnson, R.F., and Gosliner, T.M. 2018. Heterobranch Sea Slug Range Shifts in the Northeast Pacific Ocean associated with the 2015-16 El Niño. PROCEEDINGS OF THE CALIFORNIA ACADEMY OF SCIENCES 65(3): 25.

Sanford, E., Sones, J.L., García-Reyes, M., Goddard, J.H., and Largier, J.L. 2019. Widespread shifts in the coastal biota of northern California during the 2014–2016 marine heatwaves. Scientific reports 9(1): 4216.

6. The dentition data has far too limited a sample size. Given the uncertainties, as currently acknowled in the discussion, I suggest removing all reference to this aspect of the study in the methods and results. A single line in the discussion would suffice, referencing unpublished preliminary results, suggesting that the dentition information in Lindsay and Valdes may need to be re-examined with a comprehensive survey.

Minor comments

7. The first half of the first introduction paragraph (lines 51 – 59) seems irrelevant for this study.

8. The explanation regarding plankton life cycles and rafting is confusing (lines 61 – 62). Isn’t the point here that there are two possible mechanisms of dispersal for Hermissenda since it has planktonic larvae and can live in biofouling communities?

9. The paragraph on invasion and dispersal (line 86+) seems far too broad and convoluted. The key point for this study is simple: nudibranch dispersal can be via planktonic stages or via rafting.

10. Line 153 – “species” should probably be “specimens”?

11. Line 187+ – the iNaturalist methods should be in a separate paragraph

12. The first two paragraphs in the results (lines 195 – 207) do not seem to be results from this study, or at least that is what seems to be the suggestion from a related paragraph in the introduction (lines 107 -115). The information in the results seems to be extra detail from Carlton et al. publications? If they are new results, the methods by which they were achieved need to be included in the methods section. All of this information needs to more clearly explained, and only in the appropriate section (intro, methods, results, or discussion).

13. The phylogeny figures are illegible because font sizes are far too small. These raw images should be included only as supplementary figures. Alternatively, they need to be presented with changed fonts sizes, line sizes/colours and/or additional annotations to make the key points readable – that all sequences sort into two of the three clades, and they do not cause any substantive changes to the clade relationships shown by Lindsay and Valdes.

14. The bolded conclusion on line 281 seems backward. The two species are not in Japan so why would they raft? Instead, the identification of the specimens as opalescens and crassicornis means that emurai did not raft. See comment 1 above.

15. Line 319 – 320. Another case of rafting be presented as a possibility when it’s already been concluded that they did not raft. See comment 1 above.

16. Line 323+ This paragraph on diet seems mostly irrelevant. Hermissenda is a generalist that feeds in biofouling communities in its native range. Given many similar species are present in northewestern Pacific biofouling communities, it is certainly plausible that they found adequate prey supply on the debris.

17. Line 350 food preferences have already been studied:

Hoover, R.A., Armour, R., Dow, I., and Purcell, J.E. 2012. Nudibranch predation and dietary preference for the polyps of Aurelia labiata (Cnidaria: Scyphozoa). Hydrobiologia 690(1): 199–213. doi:10.1007/s10750-012-1044-x.

18. Line 355+. The information that no adult eastern Pacific species have been found on any other pieces of debris to date is compelling. This should probably be a key point in the introduction. Saving that to the very last paragraph of the discussion is counter productive.

19. The final paragraph makes an interesting point – that the alternative possibility of opalescens and crassicornis both being present in Japan is not inconsistent with the data. I suggest avoiding the tangential points regarding multiple populations and adults being washed away. Isn’t the point that if this possibility did happen, it would require one (or more) of the following: unusually short transit time from West to East, adult survival of 1+ years (despite evidence of shorter lifespans), or one or more generations, made possible by the Hermissenda living inside a tank which could have reduced dispersal by the planktonic larvae?

Reviewer #2: Montana et al. Hermissenda JTMD

Montana et al. address the origins of two Hermissenda species recovered from an approximately 8 m length of the bow of a Japanese aquaculture fish tender that arrived on the Oregon coast in 2015, four years after the 11 March 2011 Japanese earthquake and tsunami. Montana et al. use genetics and morphology to test whether these Japanese Marine Tsunami Debris (JTMD) specimens were the only known Japanese Hermissenda (H. emurai) or included other, eastern Pacific species. Their question is particularly interesting because no other mature nearshore/intertidal benthic invertebrates of the eastern Pacific were discovered on any of the other 633 JTMD items examined. Their detailed analyses, figures and tables of external morphology, life history development, SEM, biogeography and DNA sequencing are convincing. Their exposition overall is excellent and their results are original. This MS would be a worthy contribution to PLOS ONE.

Minor comments:

Montana et al. ask in their title: “Why didn’t the nudibranch cross the ocean?” but their question is “whether” the nudibranchs crossed the ocean. Their analyses reveal that only the eastern Pacific H. crassicornis and H. opalescens were present. Those Hermissenda, unknown in Japan, appear unlikely to have crossed the ocean. Their conclusion that these nudibranchs were far more likely to have settled and grown to maturity on the wreck after it reached the eastern Pacific, after it crossed the ocean, is convincing. Two other apparently eastern Pacific nudibranchs were recovered from the wreck: Dendronotus venustus and Eubranchus rustyus (visually identified) and E. rusticus confirmed also by molecular data.

Montana et al. conclude that these nudibranch pose a “striking conundrum” when considering no other of the 633 JTMD objects examined between 2012 and 2017 supported reproductively mature eastern Pacific invertebrates. More like a “challenge”? This wreck (JTMD item BF-356) was different from all other JTMD, that was launched on the afternoon of 11 March 2011. BF-356 was highly likely to have broken away from its aft section long (years) after 2011. An emphasis that the initial timing and the ocean crossing history and eastern Pacific history of BF-356 was different from all other JTMD objects would be nice to include in the analyses. Is it possible, for example, that where something drifts and lands continues to be influenced by when it was released, even years after it is released? Moreover, Montana et al. do not seem impressed that all four of the recovered nudibranchs were reproductively mature and very likely from the eastern Pacific. The exceptions were not confined to two species. Lighting struck four times, not twice. Why only nudibranchs?

General comments:

Typo at end of line 6

Typo line 364

Reviewer #3: Please, read the attached review carefully. Improvements and corrections are easy to do and they shouldn't take much time (if sequences of 18S, 16S and H3 markers are available). Some paragraph of the abstract was complicate to understand and nees to be re-written.

6. PLOS authors have the option to publish the peer review history of their article (what does this mean?). If published, this will include your full peer review and any attached files.

Reviewer #1: No

Reviewer #2: No

Reviewer #3: No

---

## [Author Response · Author response to Decision Letter 0]

20 Sep 2024

Our responses can be found in the Response to Reviewers document attached to this resubmission.

---

## [Editor Report · Decision Letter 1]

25 Sep 2024

Reversing a reported case of transoceanic dispersal: Nudibranch identifications among tsunami debris

PONE-D-24-25192R1

Dear Dr. Montana,

We’re pleased to inform you that your manuscript has been judged scientifically suitable for publication and will be formally accepted for publication once it meets all outstanding technical requirements.

Kind regards,

Luisa Borges, PhD

Academic Editor

PLOS ONE

---

## [Editor Report · Acceptance letter]

2 Oct 2024

PONE-D-24-25192R1 

PLOS ONE

Dear Dr. Montana, 

I'm pleased to inform you that your manuscript has been deemed suitable for publication in PLOS ONE. Congratulations! Your manuscript is now being handed over to our production team.

Kind regards, 

on behalf of

Dr. Luisa Borges 

Academic Editor

PLOS ONE